# Metformin regulates myoblast differentiation through an AMPK-dependent mechanism

Eleonora Maniscalco[1][☉]*, Giuliana Abbadessa[1][☉], Magalì Giordano[1], Loredana Grasso[2], Paolo Borrione[3], Silvia Racca[1]

1 Department of Clinical and Biological Sciences, University of Torino, Orbassano, Italy, 2 Department of Medical Sciences, University of Torino, Torino, Italy, 3 Department of Movement, Human and Health Sciences, University of Roma "Foro Italico", Roma, Italy

☉ These authors contributed equally to this work.
* eleonora.maniscalco@unito.it

**Data Availability Statement:** All relevant data are within the paper.

**Funding:** The authors received no specific funding for this work.

## Abstract

This study aims to investigate how metformin (Met) affects muscle tissue by evaluating the drug effects on proliferating, differentiating, and differentiated C2C12 cells. Moreover, we also investigated the role of 5'-adenosine monophosphate-activated protein kinase (AMPK) in the mechanism of action of Met. C2C12 myoblasts were cultured in growth medium with or without Met (250µM, 1mM and 10mM) for different times. Cell proliferation was evaluated by MTT assay, while cell toxicity was assessed by Trypan Blue exclusion test and Lactate Dehydrogenase release. Fluorescence Activated Cell Sorting analysis was performed to study cell cycle. Differentiating myoblasts were incubated in differentiation medium (DM) with or without 10mM Met. For experiments on myotubes, C2C12 were induced to differentiate in DM, and then treated with Met at scalar concentrations and for different times. Western blotting was performed to evaluate the expression of proteins involved in myoblast differentiation, muscle function and metabolism. In differentiating C2C12, Met inhibited cell differentiation, arrested cell cycle progression in G2/M phase and reduced the expression of cyclin-dependent kinase inhibitor 1. These effects were accompanied by activation of AMPK and modulation of the myogenic regulatory factors. Comparable results were obtained in myotubes. The use of Compound C, a specific inhibitor of AMPK, counteracted the above-mentioned Met effects. We reported that Met inhibits C2C12 differentiation probably by blocking cell-cycle progression and preventing cells permanent exit from cell-cycle. Moreover, our study provides solid evidence that most of the effects of Met on myoblasts and myotubes are mediated by AMPK.

## Introduction

Adult skeletal muscle in mammals is a stable tissue under normal circumstances but it has a remarkable ability to repair after injury. Furthermore, skeletal muscle mass may also increase in response to mechanical overload. Muscle mass growth or hypertrophy is associated with increased protein accumulation but also with proliferation and fusion of satellite cells (SCs).

**Competing interests:** The authors have declared that no competing interests exist.

SCs are tissue resident adult stem cells responsible for tissue regeneration and homeostasis. They are quiescent mononuclear precursor cells that upon activation proliferate either to become quiescent again or to differentiate and fuse with existing muscle fibres or form new myofibers. Quiescence is a reversible cell cycle arrested state characterized by the absence of cell proliferation but unlike terminally differentiated cells, quiescent cells maintain the ability to enter cell cycle and resume proliferation. To maintain the regenerative capability of muscle tissue it is imperative to keep the homeostasis of SCs and their quiescent status. Age-associated decrease in stem cell function is observed in several stem cell populations, including SCs, but the exact mechanisms involved in stem cell exhaustion are not known. In muscle, aged SCs fail to maintain quiescence and the capacity to self-renew once activated [1].

5'-adenosine monophosphate-activated protein kinase (AMPK) is a metabolic and stress sensor that in general acts to maintain cellular energy stores, by switching on the catabolic pathways that produce ATP and by switching off the anabolic pathways that consume ATP.

AMPK is heavily involved in skeletal muscle metabolic control. It has been observed that the 5-amino-4-imidazolecarboxamide ribonucleoside (AICAR), widely used to activate AMPK in various tissues, is able to activate AMPK in undifferentiated myoblasts [2]. Activation of AMPK impairs myoblasts proliferation and differentiation in culture [3]. Furthermore, transfection of C2C12 myoblasts with Ca2+/CaM-dependent protein kinase kinase β (CamKKβ), an AMPK activator, resulted in AMPK activation, cell cycle arrest and impaired proliferation as well as impaired differentiation [4, 5]. Conversely, the lack of AMPK in SCs in vivo blocks normal muscle regeneration after injury [6, 7]. Likewise, Pigna and co-workers [8] showed that treatment of colon carcinoma-bearing mice with AICAR counteracted cachexia and rescued muscle mass by recovering normal autophagy flux that is overloaded in skeletal muscle of colon carcinoma murine models. These results as whole suggest that AMPK activity is required for proper regenerative functioning, but it must be kept within tight limits. Understanding the AMPK activation role on regeneration and SCs incorporation into skeletal muscle could be useful for the development of novel therapeutic interventions to target and delay multiple mechanisms involved in skeletal muscle diseases.

Metformin (Met) has been used clinically for the treatment of type 2 diabetes since the 1960s. The principal glucose lowering effects of Met are attributed to an inhibition of hepatic glucose production and an increase in glucose utilisation in skeletal muscle [9–12]. Beyond the type 2 diabetes, Met has also been shown to have beneficial effects on multiple other disorders such as cancer, anxiety, polycystic ovary syndrome, cardiovascular and Alzheimer diseases [13–17].

As regards diseases of the musculoskeletal system, Met treatment has been shown to preserve skeletal muscle from cardiotoxin damage by protecting myotubes from necrosis without influencing muscle regeneration [18]. Furthermore, Met treatment can improve muscle function and diminish neuromuscular deficits in a murine model of Duchenne Muscular Dystrophy [19] and delay SCs activation maintaining a quiescent, low metabolic SCs state [20]. Conversely, it has been demonstrated that metformin treatment impairs muscle function through the regulation of myostatin in skeletal muscle cells via AMPK-FoxO3a-HDAC6 axis [21].

It has been suggested that AMPK may mediate many of the effects of Met, as the kinase is activated in hepatocytes and skeletal muscle after Met treatment [22, 23]. Met-induced AMPK activation can result from an increase in the AMP/ATP ratio caused by mitochondrial stress in the form of electron transport Complex I inhibition [24, 25]. However, whether AMPK is the key mediator of the effects of Met on health is uncertain.

The purpose of this study was to further investigate the effects of Met on skeletal muscle and to determine the role of AMPK in the mechanism of action of the drug. We evaluated Met

effects on proliferating, differentiating and differentiated C2C12 cells, the most commonly used cellular model to study skeletal muscle *in vitro* [26].

## Materials and methods

### Cell culture and treatments

C2C12 myoblasts were seeded on dishes at 37˚C with 5% $CO_2$ in growth medium (GM, High Glucose DMEM), supplemented with 10% foetal bovine serum, 1% penicillin-streptomycin, 4mM glutamine and 1mM sodium pyruvate.

For proliferation experiments, C2C12 myoblasts were cultured in GM and treated with or without Met at the final concentrations of 250μM, 1mM and 10mM for 24h, 48h and 72h. Met was added fresh to the medium every 24h.

To evaluate the effects of Met on cell differentiation, GM was replaced by differentiation medium (DM, 2% horse serum in High Glucose DMEM) when cells reached 80% confluence. The following 4 days, cells were treated with or without Met at the final concentration of 10mM. Met was added fresh to the medium every 24h.

For experiments on myotubes, proliferating myoblasts at ~ 80% confluence were induced to differentiate for 5–7 days by incubation in DM. The indicated concentrations of Met, Compound C (CC) or lithium chloride (LiCl) were added at the end of the differentiation process and again every 24h, when cells were provided with fresh DM. Myotubes from different conditions were harvested at the indicated time points.

### Cell proliferation assay

Cell proliferation was evaluated by using MTT colorimetric assay, in which the conversion of 3-(4,5-dimethylthiazol-2-yl)-2,5-diphenyltetrazolium bromide to insoluble tetrazolium by NAD(P)H-dependent cellular oxidoreductase was measured by recording the cell absorbance at 570 nm with a 96-well plate reader [27].

C2C12 myoblasts were seeded at 1000 cells/$cm^2$ in GM on 96-well plates. After 24h, absorbance of four plates was recorded to define the time point "0h". The remaining plates were treated with or without Met (final concentrations 250μM, 1mM, and 10mM) for 24h, 48h and 72h. Met was added fresh to the medium every 24h. Four replicates were analysed at each time point and an average of the values was calculated.

### Measurement of cell toxicity

Cell toxicity was assessed by trypan blue (TB) exclusion test and lactate dehydrogenase (LDH) release in C2C12 myoblasts exposed to different concentrations of Met (250μM, 1mM, 10mM) for 24h, 48h and 72h.

The TB exclusion assay is based on the principle that viable cells possess intact cell membranes that exclude certain dyes, such as TB. We used a commercially available TB preparation (TB solution 0.4%, Sigma-Aldrich) to perform the counts. The percentage of cell toxicity was calculated as the ratio between the number of total dead cells (stained) and total cells (stained and unstained) x100. Cells were counted by using a haemocytometer and a light microscope.

LDH release was determined in culture medium by using a photometric assay based on the conversion of pyruvic acid to lactic acid by this enzyme, in the presence of reduced NADH. Results were expressed as percentages of total LDH released by untreated cells (100%), which were lysed with PBS plus 5% Triton X-100.

## Cell cycle FACS analysis

Cell cycle was analysed in myoblasts exposed to different concentrations of Met (250µM, 1mM and 10mM) for 24h, 48h and 72h. Later, to establish if Met effects on cell cycle were reversible, C2C12 cells were exposed to 10mM Met for 24h. Subsequently, the medium was replaced with fresh medium without Met for additional 24h.

C2C12 were trypsinized, washed three times with PBS and fixed in 70% ethanol overnight at -20˚C. The cells were further washed two times with PBS and incubated with 200µl staining solution containing 0,1mg/ml RNAse, 25µg/ml propidium iodide, 0,02% Triton X-100 for 40 min at 37˚C. DNA content was measured by propidium iodide intensity by using a BD FACSVerse flow cytometer (BD Bioscience). The cell cycle phases were analysed with 162 FlowJO10.5.3.

## Immunoblotting

C2C12 myoblasts and myotubes were washed in the culture dish with ice-cold PBS and homogenized in RIPA lysis buffer (150mM NaCl, 1.0% IGEPAL® CA-630, 0.5% sodium deoxycholate, 0.1% SDS, 50mM Tris, Sigma-Aldrich) supplemented with protease inhibitor cocktail 100X (Cell Signaling). Samples were incubated in ice for 30 min with the lysis buffer and cell debris were separated by centrifugation at 14,000 rpm for 30 min at 4 ˚C. The supernatant was collected and stored at -80 ˚C. Protein concentrations were determined by Bradford colorimetric assay (Bio-Rad) [28]. Total protein extracts (30µg) were then separated by SDS-PAGE. Gels were transferred to membranes, saturated with blocking solution (5% milk and 0.1% Tween-20 in PBS), and incubated with primary antibodies overnight at 4 ˚C: rabbit anti-ACCß (1:1000, Cell Signaling 3662), rabbit anti-phospho-ACC (Ser79) antibody (1:1000, Cell Signaling 3661), rabbit anti-AMPK (1:1000, Cell Signaling 2532), rabbit anti-phospho-AMPK (Thr172) antibody (1:1000, Cell Signaling 2535), rabbit anti-cleaved-caspase-3 (1:1000, Cell Signaling 9664), mouse anti-GAPDH, (1:25000, Ambion AM4300), rabbit anti-GSK3β (1:1000, Cell Signaling 9315), rabbit anti-phospho-GSK3β (Ser9) antibody (1:1000, Cell Signaling 9323), rabbit anti-Myf5, (1:10000, Abcam ab125078), mouse anti-MYH1/2 (1:500, Santa Cruz sc-53088), mouse anti-MyoD (1:500, BD Pharmingen 554130), rabbit anti-PAX7 (1:1000, Abcam ab34360), rabbit anti-PGC1α (1:1000, Abcam ab54481), mouse anti-p21 (1:500, Santa Cruz sc-6246), mouse anti-vinculin (1:1000, Santa Cruz sc-25336).

The membranes were then washed three times and incubated with the appropriate concentrations of anti-mouse or anti-rabbit secondary antibody conjugated with horseradish peroxidase for 1h at room temperature. The blots were developed with Clarity Western ECL Substrate (Bio-Rad) using ChemiDoc™ Touch Image System (Bio-Rad). Densitometric analysis was performed using ImageLab Software. Non-phosphorylated proteins were normalized to GAPDH or vinculin. Phosphorylation level is presented as the *ratio* between phosphorylated and total protein.

## Morphological analysis

Primarily, morphological analysis was performed on myotubes treated with 10mM Met for 24h and 48h. Then, the same morphological parameters were evaluated on myotubes exposed to 10mM Met for 24h in the presence or absence of 10µM CC.

C2C12 myoblasts were seeded in six-well plates in GM supplemented with 10% foetal bovine serum, 1% penicillin-streptomycin, 4mM glutamine and 1mM sodium pyruvate. For pertinent experiments, cells were grown to ~ 80% confluence and then induced to differentiate into myotubes by incubation in DM for 5–7 days. Then myotubes were treated with 10mM

Met, 10μM CC or 10mM Met combined with 10μM CC for the appropriate times. Next, Crystal violet staining was performed, after a passage in 4% PAF to fix the cells.

Images from myotubes were visualised at X20 magnification using an inverted light microscope (Zeiss Axiovert 200) and captured with the Lumenera 3 digital camera and the Infinity Analyze 7.0 Software.

Myotube diameter and nuclei number were measured from randomly selected microscope fields from three different wells of control and treated conditions using Image J Software. No less than three diameters were measured per myotube, and at least 150 myotubes were measured per well [3].

The fusion index was calculated as the *ratio* of the nuclei number in myocytes with two or more nuclei *versus* the total number of nuclei [29]. The number of myonuclei/myotube was also evaluated.

### Statistical analysis

All data are presented as mean values±SD of at least four independent experiments. Our data showed a normal distribution according to the Kolmogorov-Smirnov test. Statistical analyses were performed with t-test or one/two-way ANOVA, followed by Bonferroni's multiple comparison test by using GraphPad Prism 8.0 software. The differences were considered significant with $^*p < 0.05$, $^{**}p < 0.01$, $^{***}p < 0.001$ and $^{****}p < 0.0001$.

## Results

### Effects of Met on proliferating C2C12

We examined the proliferation of C2C12 cells cultured in GM supplemented with different concentrations of Met (250μM, 1mM and 10mM) for 24h, 48h and 72h. Met had an anti-proliferative effect on C2C12 myoblasts. In cells exposed to 10mM Met we observed a statistically significant reduction in proliferation compared with control at all times tested. 250μM Met decreased cell proliferation only after 72h treatment, while 1mM Met reduced cell growth already starting from 48h (Fig 1).

In C2C12 exposed to the same experimental conditions, we evaluated by Western Blot (WB) analysis the effects of Met treatment on AMPK activity. A meaningful increase was found after the exposure to 1mM and 10mM Met at every time point compared with control cells (CTRL) (Fig 2).

Cell cycle arrest, cytotoxicity and apoptosis induction are the major causes of cell proliferative inhibition. To understand the mechanism by which metformin inhibited proliferation of C2C12 cells, we tested whether metformin induced cytotoxicity or apoptosis. Cell viability was monitored by the TB exclusion test and LDH release (S1 Fig). TB and LDH assays did not show statistical differences as regard the number of dead cells and cytotoxicity at any Met concentration and time point, compared with CTRL.

To elucidate whether metformin induced apoptosis WB analysis was performed to evaluate the expression of caspase-3, which plays an important role in programmed cell death. The caspase-3 expression in C2C12 cells did not vary neither in control nor in treated cells at any time point and Met concentration (S2 Fig).

At last, to establish whether Met influenced cell cycle progression, we determined the distribution of Met-treated C2C12 cells in the different phases of the cell cycle using flow cytometry at 24h (Fig 3a), 48h (Fig 3b) and 72h (Fig 3c).

The fraction of cells in G0/G1 decreased after the exposure to 10mM Met for 24h, and even more after 48h. At this time point also 1mM Met determined a significant reduction in G0/G1 cells compared with control. No differences existed in the percentage of cells in S phase. As

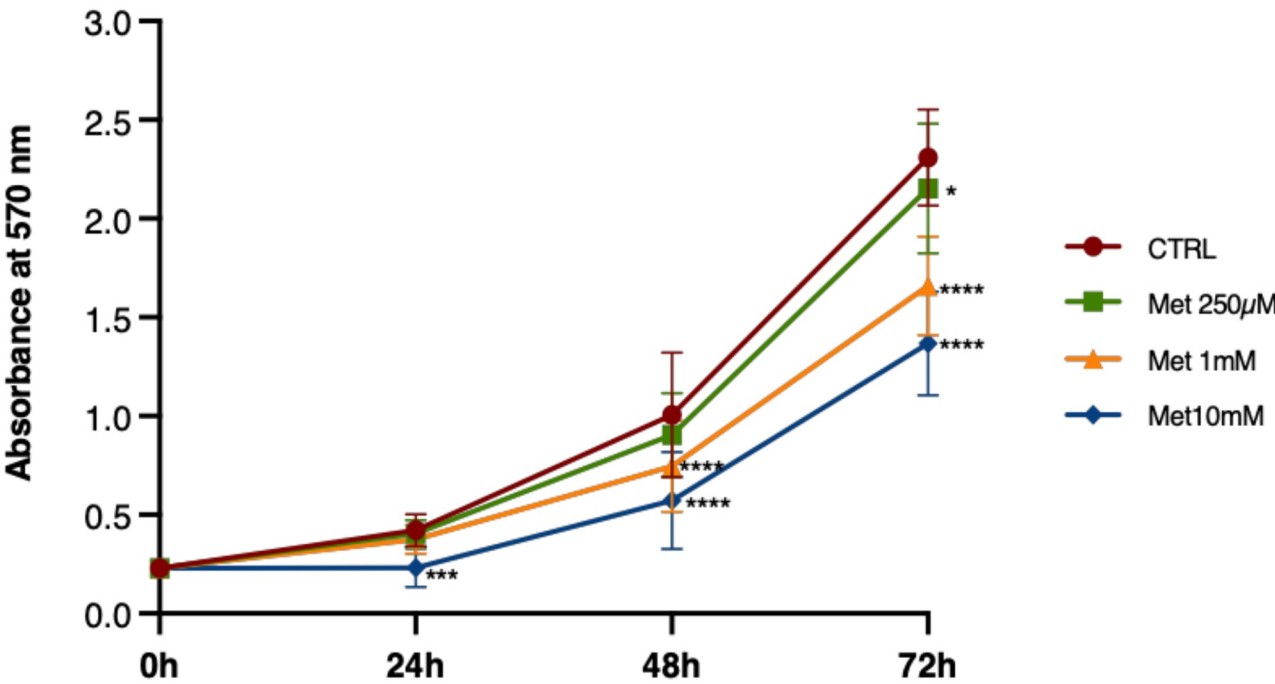

**Fig 1. Met inhibits C2C12 cell proliferation in a time- and concentration-dependent manner.** Proliferating cells were treated with 250μM, 1mM and 10mM Met for 24h, 48h and 72h. Statistical analysis was performed using two-way ANOVA followed by Bonferroni's multiple comparison test. Statistically significant differences: * ($p<0.05$), *** ($p<0.001$), **** ($p<0.0001$) *versus* CTRL.

regards cells in G2/M phase, a significant increase was found in cells treated with 1mM and 10mM Met for 24h and 48h compared with control. All the previously observed differences were undetectable at 72h.

As above reported, to establish if Met effects on cell cycle were reversible, we exposed myoblasts to 10mM Met for 24h. Subsequently, GM was replaced with fresh medium without Met and cells were incubated for an additional 24h. FACS analysis showed that Met affected cell cycle progression in an irreversible manner (Fig 3d).

## Effects of Met on myoblast differentiation

We next asked whether Met could affect myogenic differentiation process. Proliferating myoblasts were induced to differentiate by incubation in DM for a maximum of 96h and exposed to 10mM Met, added fresh to the medium every 24h.

We analysed the expression of myosin heavy chain (MYH1/2), that is considered a marker of completed differentiation, as well as the levels of the main myogenic regulatory factors (MRFs): paired box 7 (PAX7), myoblast determination protein 1 (MyoD) and myogenic factor 5 (Myf5). The expression of cyclin-dependent kinase inhibitor 1 (p21), an inhibitor of the cell cycle, was also evaluated.

MYH1/2 expression was statistically significant in CTRL only at 96 hours, while in the Met 10mM cells it was never expressed (Fig 4a and 4b).

The expression of PAX7 decreased both in CTRL and in Met10mM over time. The levels of PAX7 in treated cells were lower than the control already starting from 24h but decreased in a more attenuated manner compared with CTRL cells (Fig 4a and 4b).

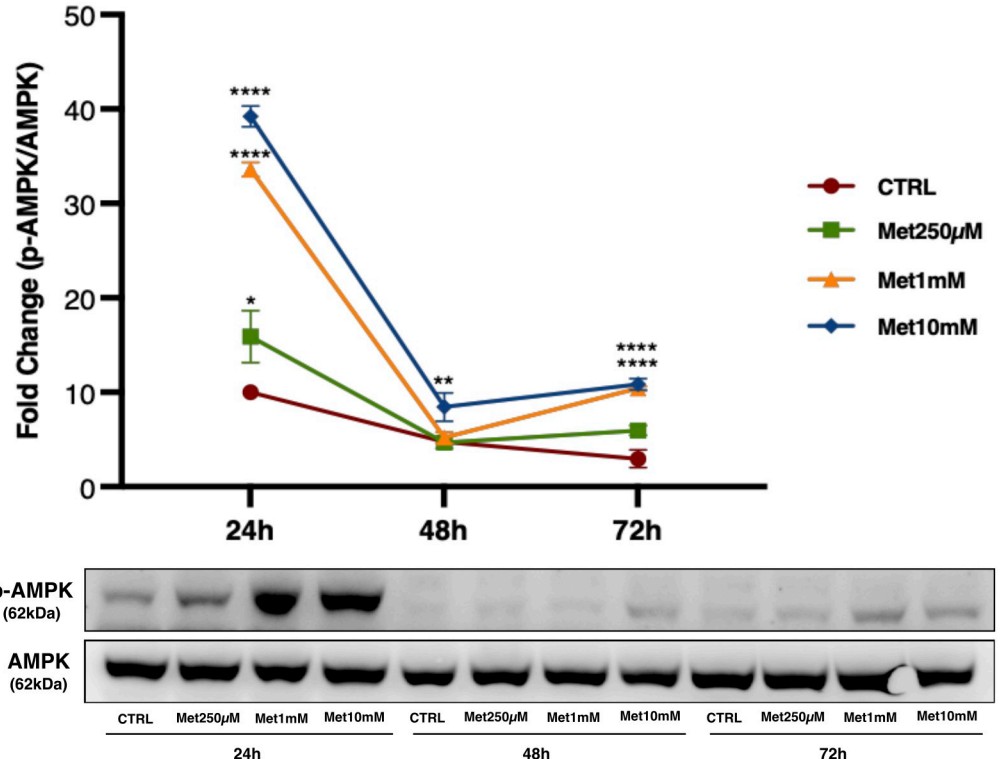

**Fig 2. Met increases AMPK activation in proliferating C2C12.** Representative image of WB for AMPK and p-AMPK in total cell lysate from C2C12 treated with 250μM, 1mM and 10mM Met for 24h, 48h and 72h. The graph represents band density expressed as a fold change of p-AMPK/AMPK *ratio* compared with CTRL from four independent experiments. * ($p<0.05$), ** ($p<0.01$), **** ($p<0.0001$) for differences *versus* CTRL using two-way ANOVA followed by Bonferroni's multiple comparison test.

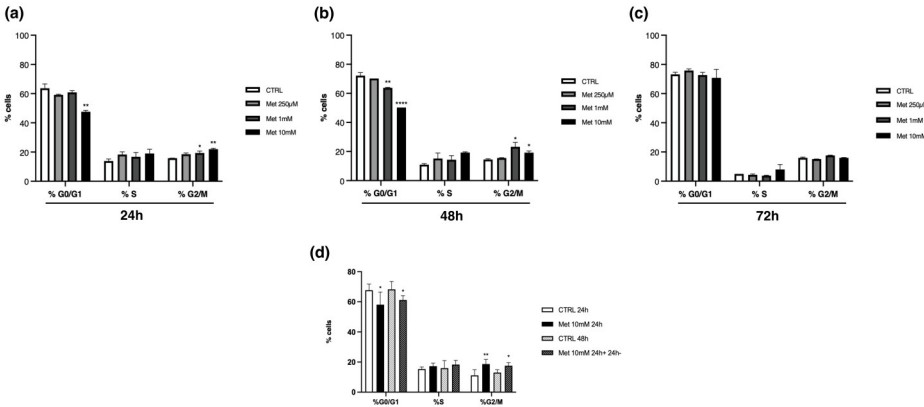

**Fig 3. Met affects C2C12 cell cycle progression.** Histograms represent flow cytometry analysis of the three different phases of the cell cycle. Cell percentage was reduced in G0/G1 phase and increased in G2/M. Statistical differences are evident at 24h (a), even more at 48h (b), and disappear at 72h (c). Statistical analysis of the differences between control and treated cells was performed for each time point and phase using one-way ANOVA followed by Bonferroni's multiple comparison test. * ($p<0.05$), ** ($p<0.01$), **** ($p<0.0001$) for differences *versus* CTRL. Cell exposure to 10mM Met for 24h and subsequently to fresh medium without Met for an additional 24h (Met 10mM 24h+ 24h-), showed that Met effect on the cell cycle lasted (d). Statistical analysis of the differences between control and treated cells was performed for each time and phase by using t-test. * ($p<0.05$), ** ($p<0.01$).

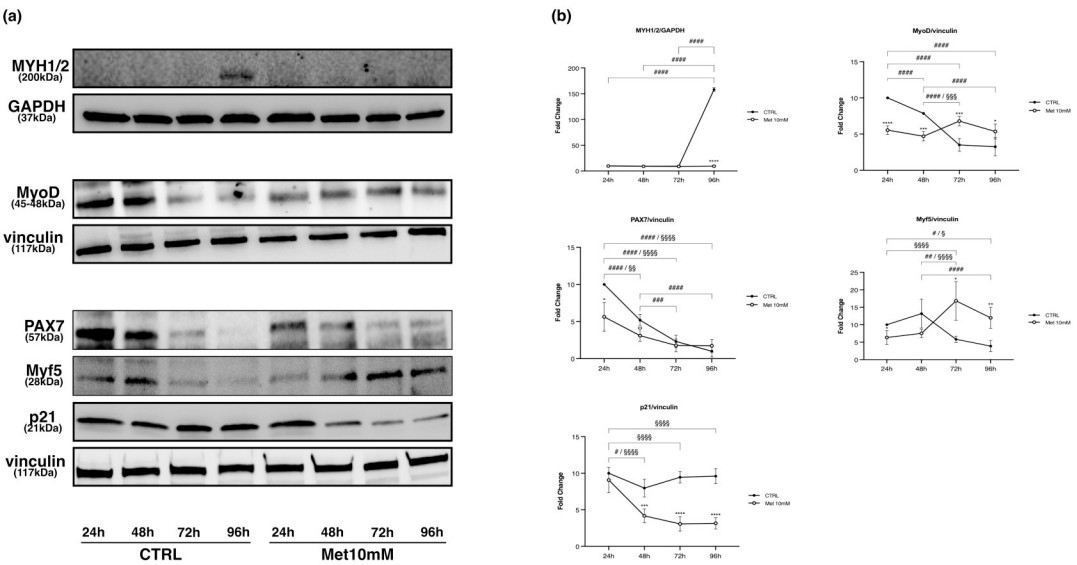

**Fig 4. Met inhibits C2C12 differentiation process.** Proliferating C2C12 cells were shifted in DM and treated with 10mM Met for 24h, 48h, 72h and 96h. (a) Representative images of WB for MYH1/2, PAX7, Myf5, MyoD and p21 in total cell lysates. (b) Graphs represent band density expressed as fold change compared with CTRL. GAPDH and vinculin were used as loading control. Data are representative of four independent experiments. Statistical analysis was performed using two-way ANOVA followed by Bonferroni's multiple comparison test. * ($p<0.05$), ** ($p<0.01$), *** ($p<0.001$), **** ($p<0.0001$) for differences between CTRL and Met; # ($p<0.05$), ## ($p<0.01$), ### ($p<0.001$), #### ($p<0.0001$) for differences within CTRL; § ($p<0.05$), §§ ($p<0.01$), §§§ ($p<0.001$), §§§§ ($p<0.0001$) for differences within Met.

Myf5 levels were lower in Met10mM cells compared with CTRL, with a significant difference at 24h. In control and treated cells, we observed an opposite trend of expression at 72h and 96h, with an increase in Met10mM and a decrease in CTRL (Fig 4a and 4b).

MyoD was meaningfully less expressed in Met10mM than in CTRL cells at 24h and 48h. While MyoD levels significantly decreased in CTRL over time, in Met10mM cells MyoD expression remained rather stable but significantly higher than control at 72h and 96h (Fig 4a and 4b).

p21 expression was always significantly lower in Met10mM compared with the CTRL cells, except for the 24h time point. In cells treated with Met, p21 meaningfully decreased over time (Fig 4a and 4b).

We also evaluated the activation of AMPK (p-AMPK/AMPK *ratio*) and the expression of factors involved in its pathway.

AMPK activation level was always significantly higher in Met10mM cells with respect to CTRL at every time point, with a tendency to decrease both in control and in Met-treated cells over time (Fig 5a).

As regards the expression of peroxisome proliferator-activated receptor-gamma coactivator 1-alpha (PGC-1α), no differences were evident between CTRL and Met10mM cells at 24h. Subsequently, PGC-1α significantly increased over time both in treated and in untreated cells, even though the increase was always lower in CTRL than Met cells. While the increase was maximal at 72h in CTRL, PGC-1α kept on rising also after 72h in Met10mM (Fig 5b).

Met induced a significant increase of acetyl-CoA carboxylase ß (ACCß) phosphorylation compared with control at each time point. Moreover, there was a significant reduction of the phosphorylation of this enzyme both in CTRL and in Met 10mM cells over time (Fig 5c).

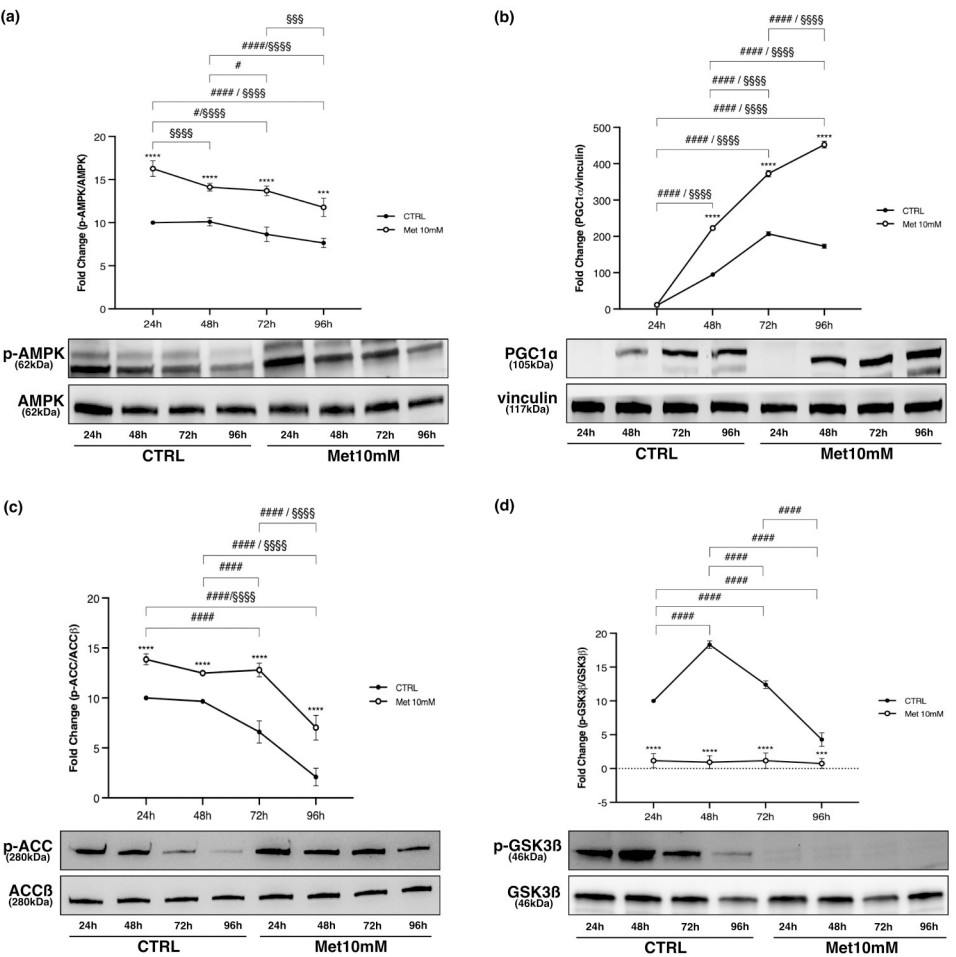

**Fig 5. Met activates AMPK, increases PGC-1α expression and ACCß phosphorylation in differentiating C2C12.**
Met maintains GSK3ß in its activated status. Proliferating C2C12 cells were shifted in DM and treated with 10mM Met for 24h, 48h, 72h and 96h. Representative images of WB for p-AMPK and AMPK (a), PGC-1α (b), p-ACC and ACCß (c), p-GSK3ß and GSK3ß (d) in total cell lysates. Graphs represent band density expressed as fold change compared with CTRL. Phosphorylation level is presented as the *ratio* between phosphorylated and total protein; PGC-1α was normalized to vinculin. Data are representative of four independent experiments. Statistical analysis was performed using two-way ANOVA followed by Bonferroni's multiple comparison test. *** ($p < 0.001$), **** ($p < 0.0001$) for differences between CTRL and Met; # ($p < 0.05$), #### ($p < 0.0001$) for differences within CTRL; §§§ ($p < 0.001$), §§§§ ($p < 0.0001$) for differences within Met.

In CTRL cells phosphorylation of glycogen synthase kinase-3ß (GSK3ß) significantly increased at 48h, it was reduced at 72h, and it plunged at 96h. In treated cells GSK3ß was not phosphorylated at any time (Fig 5d).

## Concentration-dependent effects of Met on myotubes

The levels of MYH1/2, Myf5, MyoD, p21, p-AMPK/AMPK *ratio*, PGC-1α, p-ACC/ACCß *ratio* and p-GSK3ß/GSK3ß *ratio* were evaluated in myotubes exposed to the same concentrations used in proliferating C2C12 (250μM, 1mM and 10mM Met).

The expression of MYH1/2 significantly decreased at all Met concentrations compared with CTRL. A significant reduction was also observed in Met1mM and Met10mM cells with respect to Met250μM (Fig 6a and 6b).

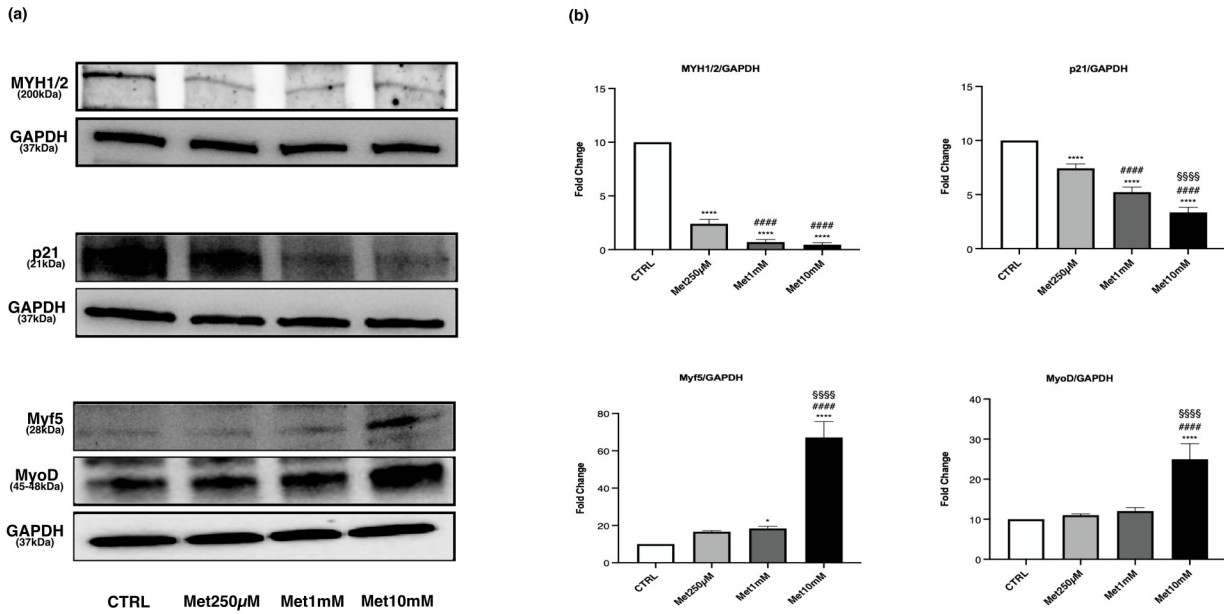

**Fig 6. Met decreases MYH1/2 and p21 expression and increases Myf5 and MyoD levels in myotubes.** Differentiated C2C12 were incubated in DM + Met at three different concentrations (250μM, 1mM, 10mM) for 24h. (a) Representative images of WB for MYH1/2, Myf5, MyoD and p21 in total cell lysates. (b) Histograms represent band density expressed as fold change compared with CTRL. GAPDH was used as loading control. Data are representative of four independent experiments. Statistical analysis was performed using one-way ANOVA followed by Bonferroni's multiple comparison test. * ($p<0.05$), **** ($p<0.0001$) for differences *versus* CTRL; #### ($p<0.0001$) for differences *versus* Met250μM; §§§§ ($p<0.0001$) for differences *versus* Met1mM.

Met treatment reduced the levels of p21 compared with control in a concentration-dependent manner (Fig 6a and 6b).

Expression levels of the myogenic factors Myf5 and MyoD were significantly higher in the cells treated with 10mM Met, compared with CTRL and cells exposed to the other two drug concentrations (Fig 6a and 6b).

1mM and 10mM Met treatment increased the phosphorylation level of AMPK compared with control and 250μM Met, even though the greatest effect on the activation of AMPK was observed after exposure to 10mM Met (Fig 7a).

The treatment with 10mM Met led to an increased expression of PGC-1α, which was statistically significant compared with CTRL, Met250μM and Met1mM cells (Fig 7b).

p-ACC/ACCß *ratio* increased in myotubes exposed to the two highest drug concentrations, with the maximal effect observed at 10mM Met (Fig 7c).

A statistically significant reduction was observed in the phosphorylation levels of GSK3ß after treatment with 1mM and 10mM Met compared with control and 250μM Met. A noteworthy decrease in p-GSK3ß/GSK3ß *ratio* was noticed in Met10mM cells (Fig 7d).

It has been reported that 400μM and 500μM Met may promote myogenic differentiation and myotube formation, whereas higher doses (1mM and 10mM) inhibit the differentiation process [30]. To verify if the effect of Met on myogenic differentiation might be related to the drug dose, we examined the same panel of factors in myotubes exposed to concentrations of Met included in the range 250μM-1mM (250μM, 400μM, 600μM, 800μM, 1mM).

Met induced a statistically significant reduction of MYH1/2 expression already starting from 400μM, with a more evident decrease at 800μM and 1mM (Fig 8a and 8b).

Compared with CTRL, the expression of the myogenic factor Myf5 significantly decreased after 400μM Met treatment and increased when myotubes were exposed to higher

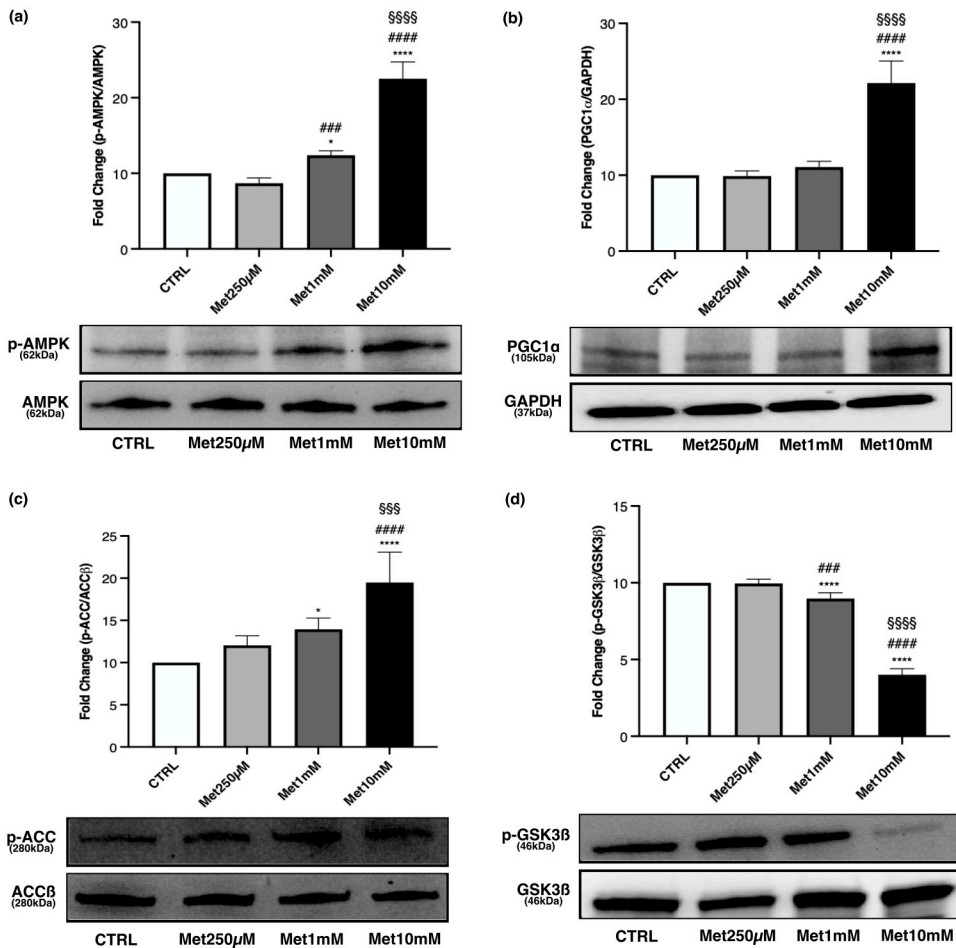

**Fig 7. Effects of Met on AMPK activation, PGC-1α expression, p-ACCß/ACCß and p-GSK3ß/GSK3ß *ratio* in myotubes.** High Met concentrations activate AMPK, increase PGC-1α expression and ACCß phosphorylation. p-GSK3ß/GSK3ß *ratio* decreases especially with 10mM Met. Differentiated C2C12 were incubated in DM + Met at three different concentrations (250μM, 1mM, 10mM) for 24h. Representative images of WB for p-AMPK and AMPK (a), PGC-1α (b), p-ACC and ACCß (c), p-GSK3ß and GSK3ß (d) in total cell lysates. Histograms represent band density expressed as fold change compared with CTRL. Phosphorylation level is presented as the *ratio* between phosphorylated and total protein; PGC-1α was normalized to GAPDH. Data are representative of four independent experiments. Statistical analysis was performed using one-way ANOVA followed by Bonferroni's multiple comparison test. * ($p<0.05$), **** ($p<0.0001$) for differences *versus* CTRL; ### ($p<0.001$), #### ($p<0.0001$) for differences *versus* Met250μM; §§§ ($p<0.001$), §§§§ ($p<0.0001$) for differences *versus* Met1mM.

concentrations, especially to 800μM (Fig 8a and 8b). Furthermore, p21 levels showed a significant reduction in treated cells with a concentration-dependent trend (Fig 8a and 8b).

Met activated AMPK starting from 400μM, and the p-AMPK/AMPK *ratio* progressively raised as the drug concentration increased (Fig 9a). As regards PGC-1α, Met treatment did not modify its expression, except for 1mM Met that induced a weak but statistically significant increase compared with CTRL (Fig 8a and 8b).

Met treatment led to a significant decrease of GSK3ß phosphorylation starting from the 400μM concentration compared with control (Fig 9b).

Morphological analysis showed that 10mM Met reduced cell diameter, fusion index, and the number of myonuclei/myotube with respect to untreated cells (Fig 10a and 10b). CC co-

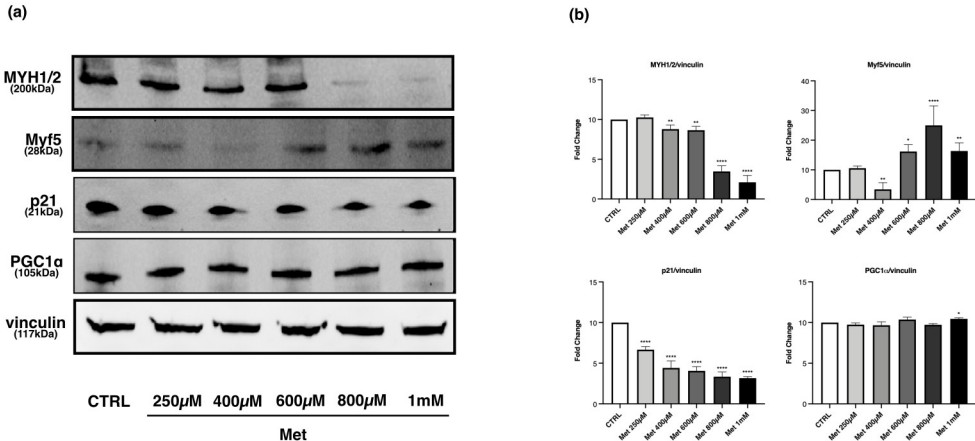

**Fig 8. Effects of Met concentrations 250μM-1mM on MYH1/2, Myf5, p21 and PGC-1α expression in myotubes.**
Differentiated C2C12 were incubated in DM + Met at the following concentrations: 250μM, 400μM, 600μM, 800μM, 1mM, for 24h. (a) Representative images of WB in total cell lysate. (b) Histograms represent band density expressed as fold change compared with CTRL. Vinculin was used as loading control. Met reduces MYH1/2 protein expression starting from 400μM, with a more evident decrease at 800μM and 1mM. The expression levels of Myf5 decreases after 400μM Met treatment and increases at higher concentrations especially at 800μM. Met reduces p21 levels in a concentration-dependent manner. Met induces a weak but statistically significant increase of PGC-1α expression only at 1mM. Data are representative of four independent experiments. Statistical analysis was performed using one-way ANOVA followed by Bonferroni's multiple comparison test. * ($p < 0.05$), ** ($p < 0.01$), **** ($p < 0.0001$) for differences *versus* CTRL.

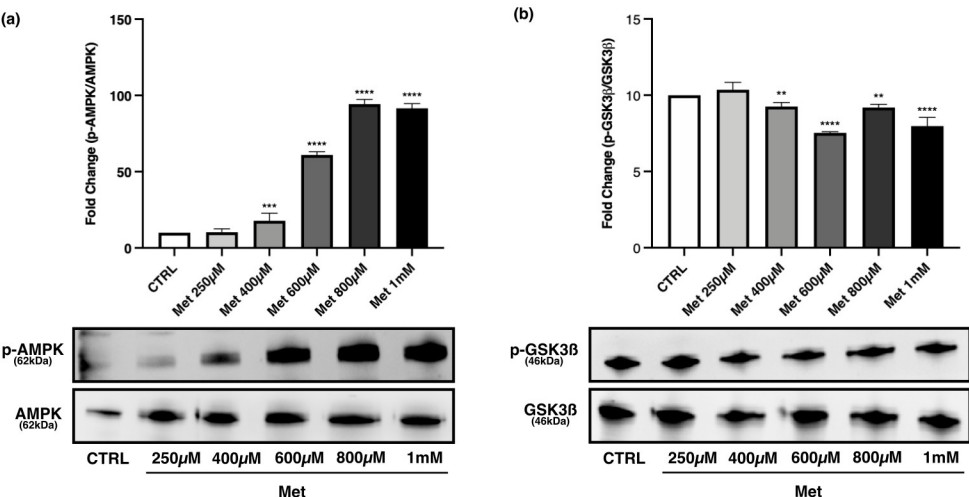

**Fig 9. Effects of Met concentrations 250μM-1mM on p-AMPK/AMPK and p-GSK3ß/GSK3ß *ratio* in myotubes.**
Differentiated C2C12 were incubated in DM + Met at the following concentrations: 250μM, 400μM, 600μM, 800μM, 1mM, for 24h. Representative images of WB in total cell lysate are shown. (a) Met activates AMPK starting from 400μM and the p-AMPK/AMPK ratio progressively raises until 800μM and remains stable at the concentration of 1mM. (b) Phosphorylation status of GSK3ß decreases starting from the exposure to 400μM Met. Histograms represent band density expressed as fold change compared with CTRL. Phosphorylation level is presented as the *ratio* between phosphorylated and total protein. Data are representative of four independent experiments. Statistical analysis was performed using one-way ANOVA followed by Bonferroni's multiple comparison test. ** ($p < 0.01$), *** ($p < 0.001$), **** ($p < 0.0001$) for differences *versus* CTRL.

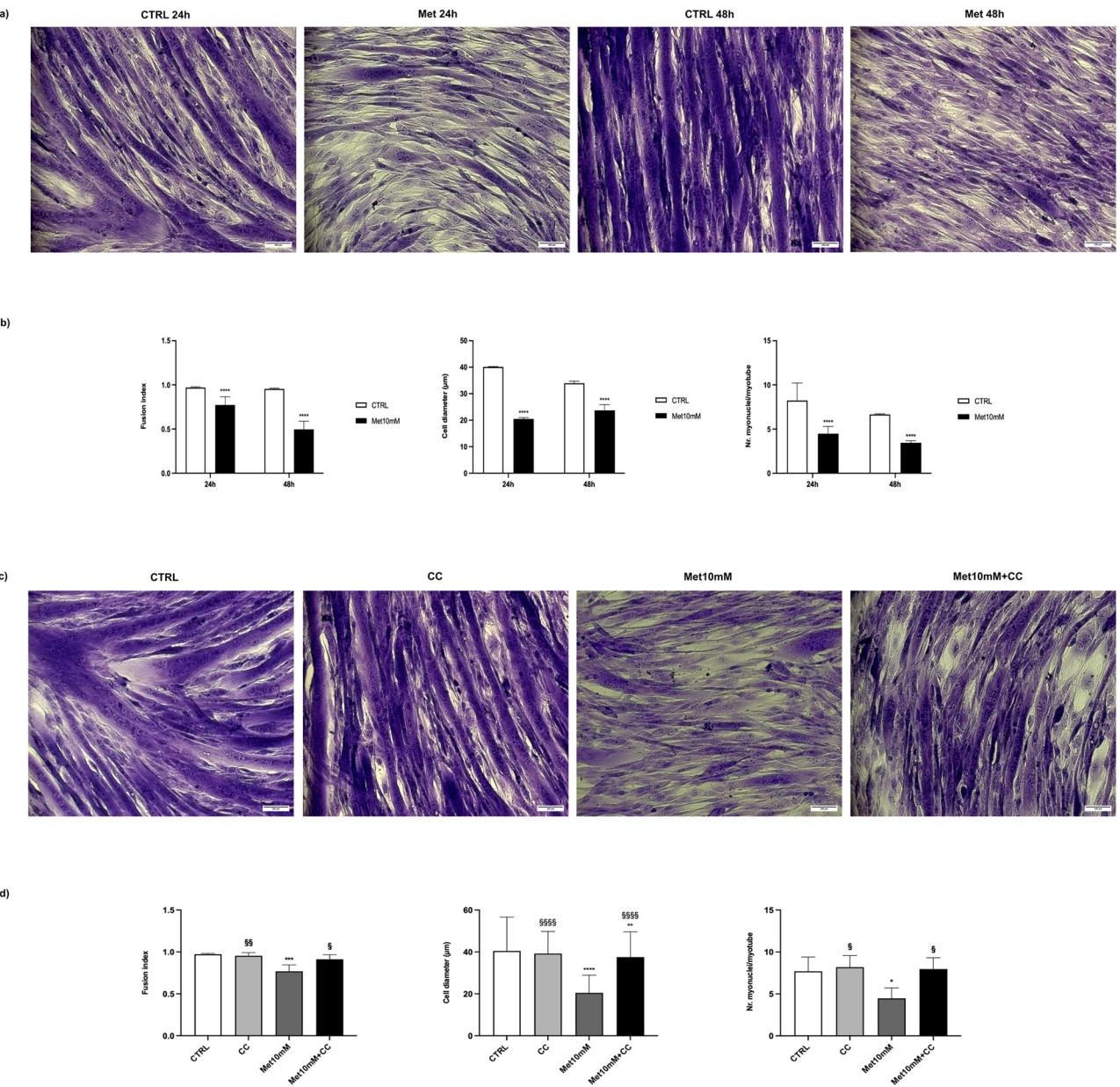

**Fig 10. Morphological analysis.** Differentiated C2C12 were incubated in DM + 10mM Met in the presence or absence of CC for 24h and with 10mM Met alone for 48h, fixed with 4% PAF and stained with Crystal violet. (a and c) Representative images of stained myotubes. (b and d) Quantification of fusion index, cell diameter and number of myonuclei/myotube. Data are representative of four independent experiments, for each one 20 random images were analysed. (b) Statistical analysis was performed using t-test. **** ($p < 0.0001$) for differences *versus* CTRL. (d) Statistical analysis was performed using one-way ANOVA followed by Bonferroni's multiple comparison test. * ($p < 0.05$), ** ($p < 0.01$), *** ($p < 0.001$), **** ($p < 0.0001$) for differences *versus* CTRL. § ($p < 0.05$), §§ ($p < 0.01$), §§§§ ($p < 0.0001$) for differences *versus* Met10mM.

administration counteracted Met effects on the abovementioned morphological parameters (Fig 10c and 10d).

## Time-dependent effects of Met on myotubes

We evaluated on myotubes the time-dependent effects of 10mM Met on the proteins previously analysed.

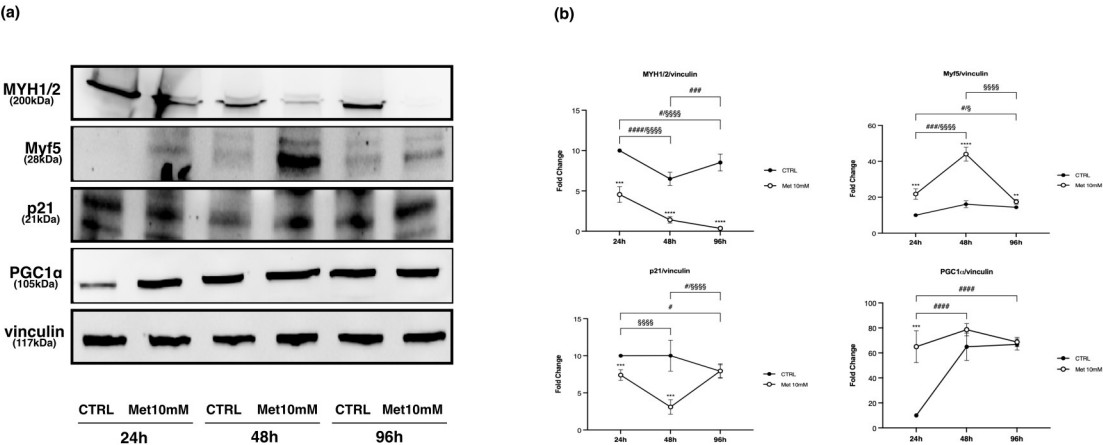

**Fig 11. Effects of Met on MYH1/2, Myf5, p21 and PGC-1α expression over time in myotubes.** Differentiated C2C12 were incubated in DM + 10mM Met for 24h, 48h and 96h. (a) Representative images of WB in total cell lysate. (b) Graphs represent band density expressed as fold change compared with CTRL. Vinculin was used as loading control. Met reduces MYH1/2 expression in a time-dependent manner. Met increases Myf5 expression with the maximal effect at 48h. Met reduces p21 levels in a time-dependent manner with the maximal effect at 48h. Met increases PGC-1α expression only at 24h. Data are representative of four independent experiments. Statistical analysis was performed using two-way ANOVA followed by Bonferroni's multiple comparison test. ** ($p < 0.01$), *** ($p < 0.001$), **** ($p < 0.0001$) for differences between CTRL and Met; # ($p < 0.05$), ### ($p < 0.001$), #### ($p < 0.0001$) for differences within CTRL; $ ($p < 0.05$), $$$$ ($p < 0.0001$) for differences within Met.

Met induced a statistically significant decrease of MYH1/2 levels in a time-dependent manner compared with control (Fig 11a and 11b).

Myf5 expression statistically increased in Met10mM cells at every time point with respect to control, with a maximal effect at 48h (Fig 11a and 11b).

p21 levels in treated cells was lower than control at 24h, and it was further reduced after 48h (Fig 11a and 11b).

PGC-1α expression significantly increased only at 24h in Met10mM cells compared with CTRL. No statistical differences were observed between treated and untreated cells at 48h and 96h (Fig 11a and 11b).

p-AMPK/AMPK *ratio* was always higher in treated myotubes compared with control at every time point, with a maximal activation at 48h (Fig 12a).

The p-GSK3ß/GSK3ß *ratio* increased in a time-dependent manner both in control and in treated cells, but the phosphorylation levels in Met10mM cells were always lower than those in CTRL with a statistically significant difference at 24h and 48h (Fig 12b).

## Cotreatment "Met-Compound C": Effects on myotubes

Compound C (CC) co-administered with 10mM Met dampened the effects of the drug on the parameters analysed.

Fig 13a and 13b show that the CC attenuated the reduction in MYH1/2 expression induced by 10mM Met. Similarly, CC counteracted the effect of Met on Myf5 expression (Fig 13a and 13b) and p-GSK3ß/GSK3ß *ratio* (Fig 13a and 13b). As regards p21 levels, cotreatment "Met-CC" not only counteracted the reduction induced by Met but led to a significant increase of the protein compared with control (Fig 13a and 13b). Furthermore, CC inhibited the upregulation of PGC-1α induced by Met (Fig 13a and 13b).

## Cotreatment "Met-lithium chloride": Effects on myotubes

Lithium chloride (LiCl) itself increased MYH1/2 levels in myotubes. The cotreatment "Met-LiCl" did not modify the effect of Met on MYH1/2 expression (Fig 14a and 14b).

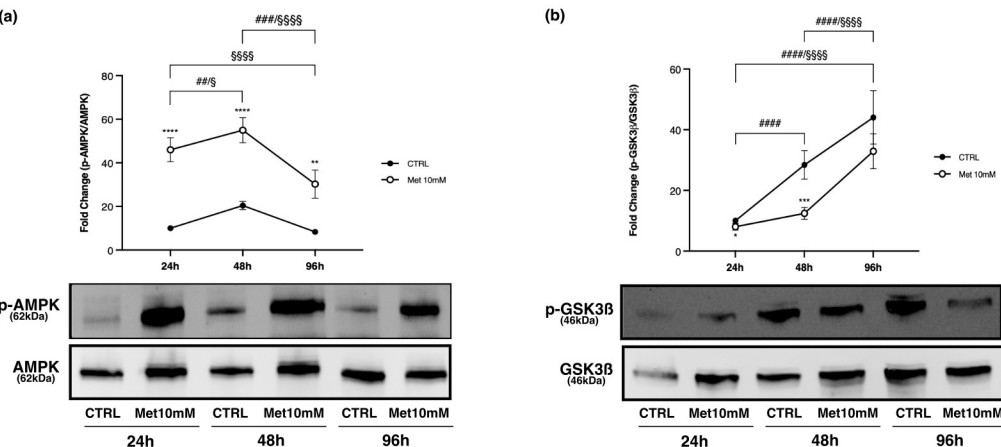

**Fig 12. Effects of Met on AMPK activation and p-GSK3ß/GSK3ß *ratio* over time in myotubes.** Differentiated C2C12 were incubated in DM + 10mM Met for 24h, 48h and 96h. Representative images of WB in total cell lysate are shown. (a) Met increases p-AMPK/AMPK *ratio* compared with CTRL. (b) p-GSK3ß/GSK3ß *ratio* increases in a time-dependent manner both in CTRL and in Met cells; GSK3ß phosphorylation was significantly lower in treated cells compared to CTRL at 24h and 48h. Graphs represent band density expressed as fold change compared with CTRL. Phosphorylation level is presented as the *ratio* between phosphorylated and total protein. Data are representative of four independent experiments. Statistical analysis was performed using two-way ANOVA followed by Bonferroni's multiple comparison test. * ($p < 0.05$), ** ($p < 0.01$), *** ($p < 0.001$), **** ($p < 0.0001$) for differences between CTRL and Met; ## ($p < 0.01$), ### ($p < 0.001$), #### ($p < 0.0001$) for differences within CTRL; § ($p < 0.05$), §§§§ ($p < 0.0001$) for differences within Met.

Met action on Myf5 and p21 was not affected by the addition of LiCl in the culture medium (Fig 14a and 14b).

A significant increase in PGC-1α was induced by LiCl alone, while the cotreatment "Met-LiCl" did not alter the effects of Met (Fig 14a and 14b).

## Discussion

The aim of the present work was to study the effects of Met on myoblast proliferation and differentiation and to understand the role of AMPK in the mechanism of action of the drug.

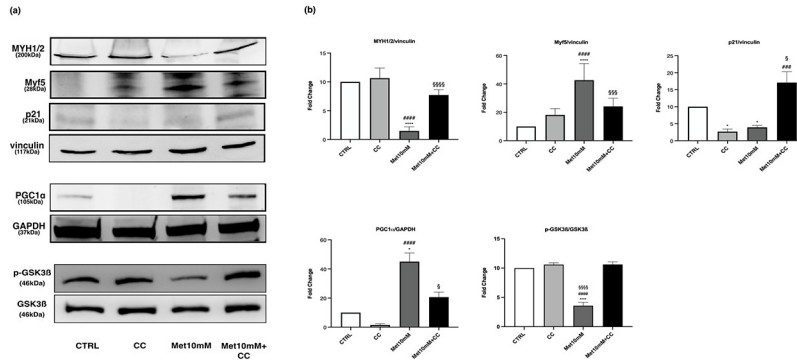

**Fig 13. CC counteracts the effects of Met on MYH1/2, Myf5, p21 and PGC-1α expression and p-GSK3ß/GSK3ß *ratio*.** Myotubes were exposed to 10mM Met alone or combined with 10μM CC for 24h. (a) Representative images of WB in total cell lysate. (b) Histograms represent band density expressed as fold change compared with CTRL. Vinculin and GAPDH were used as loading control. Data are representative of four independent experiments. Statistical analysis was performed using one-way ANOVA followed by Bonferroni's multiple comparison test. * ($p < 0.05$), **** ($p < 0.0001$) for differences *versus* CTRL, ### ($p < 0.001$), #### ($p < 0.0001$) for differences *versus* CC, § ($p < 0.05$), §§§ ($p < 0.001$), §§§§ ($p < 0.0001$) for differences *versus* Met10mM.

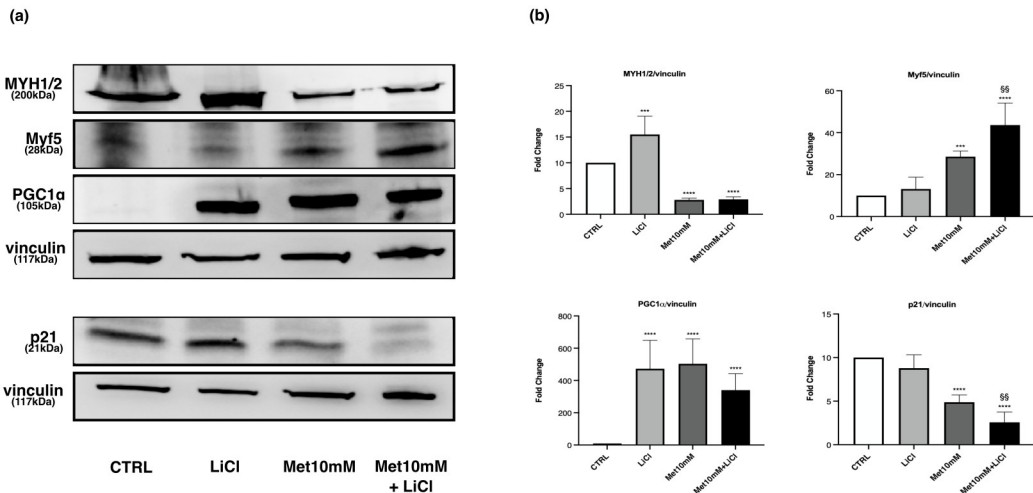

**Fig 14. LiCl does not counteract the effects of Met on MYH1/2, Myf5, PGC-1α and p21 expression.** Myotubes were exposed to 10mM Met alone or combined with 10mM LiCl for 24h. (a) Representative images of WB in total cell lysate. (b) Histograms represent band density expressed as fold change compared with CTRL. Vinculin was used as loading control. Data are representative of four independent experiments. Statistical analysis was performed using one-way ANOVA followed by Bonferroni's multiple comparison test. *** ($p<0.001$), **** ($p<0.0001$) for differences *versus* CTRL, §§ ($p<0.01$) for differences *versus* Met10mM.

We observed that Met inhibited C2C12 proliferation in a concentration- and time-dependent manner without increasing cell mortality or inducing cytotoxicity and apoptosis, according to other authors [31]. In addition, AMPK activation was detected in proliferating cells after treatment with Met. Specifically, the increase in AMPK phosphorylation was more evident in cells treated with 1mM and 10mM Met after 24h and much less, but still detectable, after 48h and 72h. The time lag observed between the increase in AMPK phosphorylation (maximal at 24h) and the inhibition of cell proliferation (maximal at 72h) suggests that Met antiproliferative effect is not a direct consequence of the AMPK activation, but some other cellular events are probably interposed between the two phenomena.

The analysis of the cell cycle showed that after treatment with 1mM and 10mM Met the percentage of myoblasts in G0/G1 phase was lower, while that of cells in G2/M was higher compared with control. These differences were evident after 24h of treatment, even more at 48h, and disappeared after 72h. We suppose that the loss of differences among control and treated cells after 72h was most likely due to the achievement of confluence in CTRL in the time span between 48h and 72h. Furthermore, since the effects of Met on the cell cycle persisted for 24h after drug removal, we assume that Met action on the cell cycle is not reversible.

Thus, our work highlights that Met-treated myoblasts decrease their proliferation rate and reside in the G2/M phase of cell cycle rather than control. These effects are associated with an increase in AMPK activation in the cells.

Williamson *et al.* [3] observed that AICAR treatment of myoblast cultures, besides increasing AMPK phosphorylation, altered cell cycle transition with a cell arrest in G1. Taken together, Williamson's and our findings suggest that the AMPK system plays a role in the regulation of cell proliferation, and we can speculate that the kinase is involved in Met effects on myoblast growth and cell cycle progression.

However, AICAR-induced AMPK activation altered cell cycle transition by inducing cell stop in G0/G1 phase, while in our study Met causes a significant increase in the percentage of

cells in G2/M. Our findings agree with those obtained by Pavlidou et al. [31] who observed a delay in cell transition through the G2/M phase after Met treatment.

In the present study, Met inhibited C2C12 myoblast differentiation as demonstrated by the lack of expression of MYH1/2, a marker of completed differentiation, in treated cells in comparison with controls in myoblasts induced to differentiate.

Moreover, a different expression pattern of the main MRFs was also evident in myoblasts induced to differentiate in presence of Met. We observed a progressive decrease of MyoD and Myf5 expression levels in differentiating CTRL. Conversely, an increase in MyoD expression and even more in Myf5 was evident in Met-treated cells over time, even though their levels were lower than controls at 24h and 48h.

It has been reported that when proliferating myoblasts are induced to differentiate by serum deprivation in the medium, a significant proportion of cells escapes from terminal differentiation. Kitzmann et al. [32] showed that MyoD and Myf5 expression patterns became mutually exclusive when C2C12 cells were induced to differentiate, with Myf5 staining present in cells which fail to differentiate. To further analyse the regulation of MyoD and Myf5 expression, authors synchronised proliferating myoblasts. Analysis of MyoD and Myf5 expression during cell cycle progression revealed that MyoD was absent in G0, peaked in mid-G1, fell to its minimum level at G1/S and made up from S to M. In contrast, Myf-5 protein was high in G0, decreased during G1 and reappeared at the end of G1 to remain stable until mitosis [32].

We can speculate that myoblasts induced to differentiate in presence of 10mM Met do not undergo a permanent exit from the cell cycle, but their cycle progression is arrested in G2/M phase. Indeed, these cells show high expression of both MyoD and Myf5 as reported by Kitzmann et al. [32] for cells from S to M phase.

The analysis of PAX7 expression profile over time in control and treated cells seems to support an alteration of the differentiation process by Met. Indeed, since PAX7 is downregulated during myogenic differentiation [33] and we observed that PAX7 downregulation was delayed by Met treatment, we may speculate that the drug could affect the differentiation process in C2C12 cells.

We then analysed the expression of p21, an inhibitor of the cell cycle. The increase in p21 level is crucial for the irreversible exit from the cell cycle and the end of the differentiation process. In differentiating cells treated with Met we observed that p21 protein expression decreased over time compared with control. p21 binds to and inhibits the activity of cyclin-CDK2, -CDK1, and -CDK4/6 complexes, and thus acts as a regulator of cell cycle progression from G1 to S phase. Moreover, in a landmark study, Spencer and co-workers [34] identified a quiescence decision point temporally located in late G2/M phase requiring high levels of p21. Cells with low p21 levels failed to enter G0 arrest.

Therefore, myoblasts treated with Met and induced to differentiate were detained in G2 probably because p21 levels were not enough to commit them to enter G0 arrest.

As already observed in proliferating myoblasts, also in differentiating cells, Met treatment induced a remarkable rise of AMPK phosphorylation.

Our results are consistent with the findings of Williamson et al. [3] showing that AICAR-treated myoblasts undergoing differentiation had a reduced myotube formation and myosin accumulation together with a reduced p21 expression.

Thus, we can suppose that Met affects myoblasts differentiation through an AMPK-dependent mechanism.

In addition, we observed that Met induced an increase in PGC-1α expression and phosphorylation of ACCß in differentiating myoblasts. ACCß is a downstream target enzyme of AMPK, and its phosphorylation promotes the oxidative metabolism in skeletal muscle. A link

between PGC-1α and AMPK is widely reported, indeed PGC-1α has been described as a mediator of some transcriptional outputs triggered by metabolic sensors like AMPK [35].

Williamson *et al.* [3] demonstrated that the activation of AMPK by AICAR in proliferating and differentiating myoblasts or in myotubes reduced p21 protein expression through a PGC-1α-dependent mechanism. Based on our data, we cannot support a direct involvement of PGC-1α in Met effects on myoblasts, however PGC-1α levels rise in differentiating cells treated with Met much more than in CTRL in the same experimental conditions.

A noteworthy result shown in the present research concerns the activation state of GSK3ß in differentiating cells. Indeed, this kinase resulted non-phosphorylated, therefore fully active in cells treated with Met at every time point, whereas its phosphorylation state was already well evident in CTRL at 24h and further increased at 48h. An important negative regulatory role for GSK3ß in myogenesis has been revealed by van der Velden and co-workers [36] who demonstrated that the promyogenic effects of Insulin-like Growth Factor-1 (IGF-1) require GSK3ß inactivation. Therefore, these data rule out the possibility that also GSK3ß could be involved in Met effects on myoblasts differentiation.

In the present study we aimed to evaluate the effects of Met also in myotube cultures. Primarily, we tested Met at the concentrations used in cell proliferation (250μM, 1mM and 10mM), then we focused on a panel of Met concentrations included in the range 250μM-1mM. Indeed, it has been reported that Met may affect myogenic differentiation with a dual action depending on the concentration employed [30]. Altogether, the effects caused by Met in myotubes were comparable to those induced in myoblasts. Since it is known that C2C12 induced to differentiate does not form myotubes with a 100% efficiency, but remains a mixed culture of myoblasts and myotubes, we hypothesise that what we observed is a consequence of the effects of Met on myoblast cell population. This speculation is supported by morphological results obtained on differentiated C2C12 exposed to Met, that show a reduction of myotubes in the mixed culture as supported by the decrease of fusion index, number of myonuclei/myotube and cell diameter.

Some of the effects on myotubes were already evident at low concentrations of Met and showed an approximative concentration-dependent trend, such as phosphorylated AMPK increase and p21 and MYH1/2 decrease. As regards GSK3ß, its phosphorylation decreased starting from 400μM Met with little ups and downs until 1mM and plunged with 10mM Met treatment. The increase in PGC-1α expression was only evident at the highest drug concentration.

Interestingly, as regards Myf5, we observed a significant reduction of its levels after 400μM Met treatment, whereas its expression increased when myotubes were exposed to higher concentrations. It is to note that Met induced AMPK activation in these cells starting from 400μM. Therefore, if we assume that the effects of Met on myotubes are mediated by an AMPK-dependent mechanism, we can speculate that 400μM Met is not enough to create cellular conditions needed to carry out the effects of drug on myotubes.

It is relevant to note that the concentrations of Met studied are higher than those achieved at therapeutic doses but exactly in line with the ones commonly used *in vitro* studies [18, 37, 38].

In addition, we showed that generally the effects of Met on myotubes reach their maximum level after 48h of treatment and tend to decrease over time, except for the PGC-1α increase. Indeed, PGC-1α expression raised in myotubes treated with Met after exposure to the highest concentration only at 24h, with no statistical difference *versus* CTRL at 48h and 96h. Therefore, Met effect on PGC-1α expression in myotubes seems to be transient and to require high concentrations of the drug. The lack of statistically significant difference between control and treated cells both at 48h and 96h is due to an increased PGC-1α expression also in CTRL at

48h, probably correlated with the increment of AMPK phosphorylation observed in these cells at the same time point.

As we underlined above, when proliferating or differentiating myoblasts or myotubes were treated with Met, a significant increase of AMPK activation was evident. In each of these cell types, Met-induced activation of AMPK was accompanied by a series of other cellular events that have already been partially demonstrated in the same cell types after AMPK activation by AICAR treatment [3].

Based on these analogies, we hypothesized that Met effects in skeletal muscle cells are mediated by AMPK. Moreover, our findings showed that GSK3ß could also be involved in Met inhibition of myoblast differentiation.

Therefore, to test the contribution of AMPK or GSK3ß to the effects of Met on myotubes, we evaluated how Met affected the expression of the parameters mainly involved in myoblast differentiation when a specific inhibitor of AMPK or GSK3ß was added in culture.

## Conclusions

In this work we studied the effects of Met on myoblasts proliferation and differentiation and determined the role of AMPK in the mechanism of action of the drug.

We observed that AMPK inhibition by CC attenuated the effects of Met on the measured morphological parameters and on the expression of all the analysed proteins, supporting the hypothesis that Met impairs skeletal muscle differentiation through an AMPK-dependent mechanism. Interestingly, CC counteracted the effect of Met on p-GSK3ß/GSK3ß *ratio*, revealing an AMPK-mediated effect of Met on activated-GSK3ß status, although our data do not allow us to establish the exact mechanism by which AMPK affects GSK3ß status. To investigate the role of GSK3ß in Met effects on myotubes, we employed LiCl, a specific pharmacological inhibitor of this kinase [39]. The addition of LiCl in the culture medium of treated myotubes did not counteract the action of Met on p21 and Myf5 expression. As regards MYH1/2, LiCl did not change the effects of Met, nevertheless LiCl alone increased MYH1/2 expression. Inhibition of GSK3ß by LiCl caused an augment of PGC-1α comparable to that induced by Met. This result is consistent with reports that connect knock-down of GSK3ß, as well as its pharmacological inhibition, to increased PGC-1α transcript and protein abundance during myogenic differentiation and in fully differentiated myotubes [40, 41]. As Met and LiCl increased PGC-1α expression themselves compared with control, but their association did not lead to an additional increase in PGC-1α, we may speculate that a common mechanism is involved in their final effect on PGC-1α expression, even though this hypothesis needs to be further investigated.

In conclusion, we observed that high Met concentrations decreased myoblasts proliferation rate without inducing cell toxicity and apoptosis. Moreover, Met inhibited C2C12 differentiation probably by blocking cell-cycle progression in G2/M phase and preventing cells permanent exit from cell-cycle and their postmitotic fusion with adjacent myoblasts. These effects were accompanied by AMPK activation observed both in proliferating and in differentiating myoblasts and in myotubes. Our findings are consistent with reports obtained in C2C12 and satellite cells *in vitro* [20, 31] and inconsistent with other results achieved *in vivo* and *in vitro* [42–44].

Though the effects of Met on myogenic differentiation are still controversial and further investigations are needed, our study provides solid evidence that most of the effects of Met on myoblasts and myotubes are mediated by AMPK.

## Supporting information

**S1 Fig. Met does not induce cytotoxicity in proliferating C2C12 cells.** TB exclusion test (a) and LDH release assessment (b) were performed. Statistical analysis of the differences between control and treated cells was conducted for each time point using one-way ANOVA. No differences were observed ($p > 0.05$).
(TIF)

**S2 Fig. Met does not induce apoptosis in proliferating C2C12 cells.** WB analysis revealed no differences in caspase-3 expression in Met treated myoblast at any time point and concentration.
(TIF)

**S1 Raw images.**
(PDF)

## Author Contributions

**Conceptualization:** Silvia Racca.

**Data curation:** Eleonora Maniscalco, Giuliana Abbadessa.

**Formal analysis:** Giuliana Abbadessa.

**Investigation:** Eleonora Maniscalco, Giuliana Abbadessa, Magalì Giordano.

**Resources:** Silvia Racca.

**Software:** Eleonora Maniscalco.

**Supervision:** Giuliana Abbadessa, Silvia Racca.

**Writing – original draft:** Eleonora Maniscalco, Giuliana Abbadessa, Loredana Grasso, Paolo Borrione, Silvia Racca.

**Writing – review & editing:** Eleonora Maniscalco, Giuliana Abbadessa, Silvia Racca.

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
