## [Decision Letter · Decision Letter 0]

14 Jul 2022

PONE-D-22-17370Metformin regulates myoblast differentiation through an AMPK-dependent mechanismPLOS ONE

Dear Dr. Maniscalco,

Thank you for submitting your manuscript to PLOS ONE. After careful consideration, we feel that it has merit but does not fully meet PLOS ONE’s publication criteria as it currently stands. Therefore, we invite you to submit a revised version of the manuscript that addresses all the points raised during the review process. In particular, the reviewers notified that most of the observed results have been already published in Pavlidou et al. 2017 (10.1371/journal.pone.0182475), thus limiting the originality of the study. The novel part of the study, as also stated in the title of the manuscript, is that Metformin regulates myoblast differentiation through an AMPK-dependent mechanism. Nevertheless, the reported data do not support the conclusions. Thus, you are invited to provide solid data to support the conclusions that metformin activity is AMPK-dependent.  

We look forward to receiving your revised manuscript.

Kind regards,

Antonio Musaro, Ph.D.

Academic Editor

PLOS ONE

Journal Requirements:

Reviewers' comments:

Reviewer's Responses to Questions

**Comments to the Author**

1. Is the manuscript technically sound, and do the data support the conclusions?

Reviewer #1: Partly

Reviewer #2: Yes

2. Has the statistical analysis been performed appropriately and rigorously? 

Reviewer #1: Yes

Reviewer #2: Yes

3. Have the authors made all data underlying the findings in their manuscript fully available?

Reviewer #1: Yes

Reviewer #2: Yes

4. Is the manuscript presented in an intelligible fashion and written in standard English?

Reviewer #1: Yes

Reviewer #2: Yes

5. Review Comments to the Author

Reviewer #1: In this manuscript, Maniscalco and colleagues explore the effects of Metformin treatment on myogenic differentiation of C2C12 (a murine myoblast cell line). The authors convincingly show an alteration of myoblast proliferation and differentiation using proliferation/cell death assays and western blot. I was very surprised by the total absence of cell culture images to correlate molecular evidence with histological staining and morphological analysis. Moreover, most of the observed results have already been published in Pavlidou et al. 2017 (10.1371/journal.pone.0182475).

Finally, the experiment with the AMPK inhibitor was very poor (one figure and limited to 4 WB) and does not support the conclusion that metformin activity is AMPK-dependent. I suggest that the authors extend AMPK inhibitor treatment to all biological observations described in the manuscript to increase the novelty of this work. For these reasons, I consider that the manuscript is not ready for publication on this journal.

Reviewer #2: The manuscript by Maniscalco et al. aims to show the effects of Metformin (Met) on myoblasts proliferation and differentiation by exploiting an in vitro model (C2C12 cells). The authors show a dose- and time-dependent effect of Met on C2C12 cell differentiation: overall the result indicate that Met decreases differentiation, by keeping myoblasts in an activated state, characterized by proliferation and increase in MyoD and Myf5, while decreasing the markers of terminal differentiation. This is associated with AMPK activation. By combining observational experiments (WB for AMPK) and pharmacological treatments (Met compound C and Li) the authors demonstrate the involvement of AMPK in mediating Met effects.

The study of Metformin biological effects I relevant, since this compound is already used in the clinical practice. The experiments are sound and well designed, even though the in vitro model limits the relevance of the findings.

MAJOR REMARKS

- Are met effects reversible? C2C12 are a satellite cell-derived cel line and should retain some features of satellite cells, including the capability to temporarily exit from the cell cycle and then re-enter the cell cycle when they are “activated” and become myoblasts, which ultimately fuse into myotubes.

- Results, figures 7 to 9 etc... The authors present the results as regarding “myotubes”, while what they measured is likely happening in mixed culture of myoblasts and myotubes, since it is known that C2C12 do not form myotubes with a 100% efficiency. It would be very informative to show morphometric analyses of these cultures and the quantification of different morphological parameters of myogenic differentiation, such as fusion index (% of cells fused into myotubes), number of myonuclei/myotube etc. This would allow to: 1) confirm with a direct approach the effects of Met on myogenic cells; 2) help to interpret how much of the WB results is accounted for by the effects on myoblasts vs the effects on myotubes.

- The authors should consider another in vivo study on AMPK activation, that is the capacity to counteract muscle atrophy induced by a tumor, i.e. cancer cachexia (Pigna et al 2016). Citing this paper would further highlight that the results in vitro and in vivo may differ significantly and the AMPK role in myogenic differentiation could be very different from that in muscle homeostasis. The discussion of this paper could be placed either in the introduction (around lines 73-74, references 6 and 7, or around lines 84-90, refs 17-20) or in the discussion.

- Statistical analysis. Since parametric analysis (ANOVA) was used, I assume that the samples had a normal distribution. Please, report the name of the test used to verify this and the other conditions to perform parametric analysis, along with the results of the test.

In particular, on Figure 1 statistics. Why using one-way ANOVA followed by Bonferroni’s multiple comparison test? In order to claim that “Met inhibits C2C12 cell proliferation in a time- and concentration-dependent manner” a two-way-ANOVA or even a MANOVA would be more appropriate. Also, the asterisks within the graph would help to localize the specific, statistical differences.

MINOR REMARKS

- results, figure 3. These negative results could be presented as Supplemental material to help the flow of the presentation.

- Abstract , line 25; introduction, line 96. The author cannot state that their study aims to invertigate muscle physiology, while presenting data obtained in vitro. Please, change the word or rephrase

- related to the remark above, I have a comment on the Metformin concentrations. Those in the microM range are much more closer to those having an effect in vivo, i.e. the “physiological ones”, while the mM concentrations appear very high. Can the authors comment/justify their choices throughout the text?

- Abstract, line 27; intro, line 97. Why “actual”? Just “the role of” is OK.

6. PLOS authors have the option to publish the peer review history of their article (what does this mean?). If published, this will include your full peer review and any attached files.

Reviewer #1: No

Reviewer #2: **Yes: **Dario Coletti

---

## [Author Response · Author response to Decision Letter 0]

22 Dec 2022

Authors’ responses to Reviewers:

Reviewer #1 

Comment to the Authors: In this manuscript, Maniscalco and colleagues explore the effects of Metformin treatment on myogenic differentiation of C2C12 (a murine myoblast cell line). The authors convincingly show an alteration of myoblast proliferation and differentiation using proliferation/cell death assays and western blot. I was very surprised by the total absence of cell culture images to correlate molecular evidence with histological staining and morphological analysis. Moreover, most of the observed results have already been published in Pavlidou et al. 2017 (10.1371/journal.pone.0182475).

Finally, the experiment with the AMPK inhibitor was very poor (one figure and limited to 4 WB) and does not support the conclusion that metformin activity is AMPK-dependent. I suggest that the authors extend AMPK inhibitor treatment to all biological observations described in the manuscript to increase the novelty of this work. For these reasons, I consider that the manuscript is not ready for publication in this journal.

Authors’ response/action: We are grateful to the Reviewer for arising this remark and we agree with him with the need to carry out histological staining and morphological analysis to correlate with molecular data. These analyses on myotubes have been inserted in the manuscript (Materials and methods, lines 183-196; Results, lines 404-405, Fig.10; Discussion, lines 580-585).

The reviewer considers the experiment with the AMPK inhibitor very poor and that it does not support the conclusion that metformin activity is AMPK-dependent. 

We limited our analyses with compound C to the parameters that, among those studied, we thought were most involved in the control of differentiation such as Myf5, p21, PGC1-alpha as well as MYH1/2, marker of terminal differentiation.

We showed that Met decreases C2C12 differentiation by keeping myoblasts in an activated state, characterised by an increase in MyoD and Myf5 and a decrease of MYH1/2. This is associated with AMPK activation. Since we demonstrated that the treatment with Compound C, an inhibitor of AMPK, counteracted the effects of Met on the above-mentioned parameters, we think we sufficiently demonstrated the involvement of AMPK in mediating Met effects.

Nevertheless, we accepted the reviewer’s suggestion, and we evaluated the effect of Compound C alone or in combination with Met on p-GSK3ß/GSK3ß ratio, an enzyme involved in myoblasts differentiation (Results, lines 456-457, Fig 13c; Conclusions, lines 623-626).

Reviewer #2:

Comment to the Authors: The manuscript by Maniscalco et al. aims to show the effects of Metformin (Met) on myoblasts proliferation and differentiation by exploiting an in vitro model (C2C12 cells). The authors show a dose- and time-dependent effect of Met on C2C12 cell differentiation: overall the result indicate that Met decreases differentiation, by keeping myoblasts in an activated state, characterised by proliferation and increase in MyoD and Myf5, while decreasing the markers of terminal differentiation. This is associated with AMPK activation. By combining observational experiments (WB for AMPK) and pharmacological treatments (Met compound C and Li) the authors demonstrate the involvement of AMPK in mediating Met effects.

The study of Metformin biological effects is relevant, since this compound is already used in clinical practice. The experiments are sound and well designed, even though the in vitro model limits the relevance of the findings.

Authors’ response/action: We thank the Reviewer for his comments and for appreciating the design of the study.

MAJOR REMARKS

- Are Met effects reversible? C2C12 are a satellite-derived cell line and should retain some features of satellite cells, including the capability to temporarily exit from the cell cycle and then re-enter the cell cycle when they are “activated” and become myoblasts, which ultimately fuse into myotubes.

Authors’ response/action: To answer this question the effects of Met on cell cycle was further investigated. For this purpose, C2C12 cells were exposed to 10mM Met for 24h. Subsequently, the medium was replaced with fresh medium without Met for additional 24h and the effects were analysed by flow cytometer (Materials and methods, lines 149-151). The results obtained suggest that Met effects are not reversible (Results, lines 257-260; Discussion, lines 501-502).

- Results, figures 7 to 9 etc... The authors present the results as regarding “myotubes”, while what they measured is likely happening in a mixed culture of myoblasts and myotubes, since it is known that C2C12 do not form myotubes with a 100% efficiency. It would be very informative to show morphometric analyses of these cultures and the quantification of different morphological parameters of myogenic differentiation, such as fusion index (% of cells fused into myotubes), number of myonuclei/myotube etc. This would allow to: 1) confirm with a direct approach the effects of Met on myogenic cells; 2) help to interpret how much of the WB results is accounted for by the effects on myoblasts vs the effects on myotubes.

Authors’ response/action: We thank the reviewer for his precious suggestion and we agree with him on the importance of performing morphometric analyses of our cultures to quantify different morphological parameters of myogenic differentiation and to interpret how much of the WB results is accounted for by the effects on myoblasts vs the effects on myotubes (Materials and methods, lines 183-196). 

The morphological results obtained in differentiated C2C12 exposed to Met showed a decrease of fusion index, number of myonuclei/myotube and cell diameter (Results, lines 404-405, Fig 10). These data suggest that what we observed in myotubes is a consequence of the effects of Met mainly on myoblast cell population (Discussion, lines 580-585).

- The authors should consider another in vivo study on AMPK activation, that is the capacity to counteract muscle atrophy induced by a tumor, i.e. cancer cachexia (Pigna et al. 2016). Citing this paper would further highlight that the results in vitro and in vivo may differ significantly and the AMPK role in myogenic differentiation could be very different from that in muscle homeostasis. The discussion of this paper could be placed either in the introduction (around lines 73-74, references 6 and 7, or around lines 84-90, refs 17-20) or in the discussion.

Authors’ response/action: We accepted the reviewer’s suggestion. We cited the paper by Pigna et al and we placed it in the introduction (Introduction, lines 75-77).

- Statistical analysis. Since parametric analysis (ANOVA) was used, I assume that the samples had a normal distribution. Please, report the name of the test used to verify this and the other conditions to perform parametric analysis, along with the results of the test.

In particular, on Figure 1 statistics. Why using one-way ANOVA followed by Bonferroni’s multiple comparison test? In order to claim that “Met inhibits C2C12 cell proliferation in a time- and concentration-dependent manner” a two-way-ANOVA or even a MANOVA would be more appropriate. Also, the asterisks within the graph would help to localise the specific, statistical differences.

Authors’ response/action: We thank the referee for the observations. We reported the name of the test used to verify samples' normal distribution (Statistical Analysis, lines 199-200). As regards Fig.1, we apologise for the typing error: we performed two-way-ANOVA but in the text we wrongly typed “one” instead of “two”. We placed the asterisks within the graph, as suggested.

MINOR REMARKS

- results, figure 3. These negative results could be presented as Supplemental material to help the flow of the presentation.

Authors’ response/action: We did it.

- Abstract, line 25; introduction, line 96. The author cannot state that their study aims to investigate muscle physiology, while presenting data obtained in vitro. Please, change the word or rephrase.

Authors’ response/action: We did it.

- related to the remark above, I have a comment on the Metformin concentrations. Those in the microM range are much more closer to those having an effect in vivo, i.e. the “physiological ones”,

while the mM concentrations appear very high. Can the authors comment/justify their choices throughout the text?

Authors’ response/action: The concentrations of Met studied are higher than those achieved at therapeutic doses but exactly in line with the ones commonly used in vitro studies (Discussion, lines 597-598; References, 18,36,37).

- Abstract, line 27; intro, line 97. Why “actual”? Just “the role of” is OK.

Authors’ response/action: We did it.

---

## [Decision Letter · Decision Letter 1]

27 Dec 2022

PONE-D-22-17370R1Metformin regulates myoblast differentiation through an AMPK-dependent mechanismPLOS ONE

Dear Dr. Maniscalco,

Thank you for submitting your manuscript to PLOS ONE. After careful consideration, we feel that it has merit but does not fully meet PLOS ONE’s publication criteria as it currently stands. Therefore, we invite you to take into account the additional comments of the reviewer and submit a revised version of the manuscript that addresses the points raised during the review process.

We look forward to receiving your revised manuscript.

Kind regards,

Antonio Musaro, Ph.D.

Academic Editor

PLOS ONE

Journal Requirements:

Reviewers' comments:

Reviewer's Responses to Questions

**Comments to the Author**

1. If the authors have adequately addressed your comments raised in a previous round of review and you feel that this manuscript is now acceptable for publication, you may indicate that here to bypass the “Comments to the Author” section, enter your conflict of interest statement in the “Confidential to Editor” section, and submit your "Accept" recommendation.

Reviewer #1: (No Response)

2. Is the manuscript technically sound, and do the data support the conclusions?

Reviewer #1: Yes

3. Has the statistical analysis been performed appropriately and rigorously? 

Reviewer #1: Yes

4. Have the authors made all data underlying the findings in their manuscript fully available?

Reviewer #1: Yes

5. Is the manuscript presented in an intelligible fashion and written in standard English?

Reviewer #1: Yes

6. Review Comments to the Author

Reviewer #1: The manuscript is improved over the previous version. However, the authors need to show whether the molecular effects of Ampk inhibitor treatment are reflected in a recovery of myotubes morphology as well. So the authors need to add the analyses shown in Figure 10 also for myotubes treated with compound C.

The housekeeping bands shown in Figure 4b-4c and 4e are the same. The authors should eliminate duplicates and show in one panel the bands revealed on the same filter. The same should be done for the blots in Figure 6b-6c, Figure 6a-6d, Figure 8a-8b-8c-9b, Figure 9a-9c, Figure 11a-11b-11c-12b, Figure 13a-13b-13d, Figure 14a-14b-14d.

7. PLOS authors have the option to publish the peer review history of their article (what does this mean?). If published, this will include your full peer review and any attached files.

Reviewer #1: No

---

## [Author Response · Author response to Decision Letter 1]

30 Jan 2023

Dear Editor,

thank you for the comments regarding our manuscript (PONE-D-17370R1) entitled “Metformin regulates myoblast differentiation through an AMPK-dependent mechanism” by E. Maniscalco et al. 

The Reviewer’s remarks spurred us to further improve the overall quality of our research.

Yours faithfully,

Eleonora Maniscalco

Authors’ responses to Reviewer #1:

Reviewer #1 

Comment to the Authors: The manuscript is improved over the previous version. However, the authors need to show whether the molecular effects of Ampk inhibitor treatment are reflected in a recovery of myotubes morphology as well. So the authors need to add the analyses shown in Figure 10 also for myotubes treated with compound C. The housekeeping bands shown in Figure 4b-4c and 4e are the same. The authors should eliminate duplicates and show in one panel the bands revealed on the same filter. The same should be done for the blots in Figure 6b-6c, Figure 6a-6d, Figure 8a-8b-8c-9b, Figure 9a-9c, Figure 11a-11b-11c-12b, Figure 13a-13b-13d, Figure 14a-14b-14d.

Authors’ response/action: We agree with the Reviewer’s remarks.

As suggested, we evaluated the effects of the AMPK inhibitor on myotubes at a morphological level. We observed that the molecular effects of compound C are reflected in a recovery of myotubes morphology as well. These analyses have been inserted in the manuscript (Materials and methods, lines 184-186; 190-191; Results, lines 408-409, Figs 10c and 10d; Conclusions, lines 629-630).

As regards the remarks on blot figures, we eliminated the housekeeping duplicates and showed together the bands revealed on the same filter according to the Reviewer’s suggestion (Fig 4, Fig 6, Fig 8, Fig 9, Fig 11, Fig 12, Fig 13, Fig 14 and related captions).

---

## [Decision Letter · Decision Letter 2]

31 Jan 2023

Metformin regulates myoblast differentiation through an AMPK-dependent mechanism

PONE-D-22-17370R2

Dear Dr. Maniscalco,

We’re pleased to inform you that your manuscript has been judged scientifically suitable for publication and will be formally accepted for publication once it meets all outstanding technical requirements.

Kind regards,

Antonio Musarò, Ph.D.

Academic Editor

PLOS ONE

Additional Editor Comments (optional):

Reviewers' comments:

Reviewer's Responses to Questions

**Comments to the Author**

1. If the authors have adequately addressed your comments raised in a previous round of review and you feel that this manuscript is now acceptable for publication, you may indicate that here to bypass the “Comments to the Author” section, enter your conflict of interest statement in the “Confidential to Editor” section, and submit your "Accept" recommendation.

Reviewer #1: All comments have been addressed

2. Is the manuscript technically sound, and do the data support the conclusions?

Reviewer #1: Yes

3. Has the statistical analysis been performed appropriately and rigorously? 

Reviewer #1: Yes

4. Have the authors made all data underlying the findings in their manuscript fully available?

Reviewer #1: Yes

5. Is the manuscript presented in an intelligible fashion and written in standard English?

Reviewer #1: Yes

6. Review Comments to the Author

Reviewer #1: The authors fully accomplish the referee’s queries and the revised version of the manuscript as been improved. The manuscript is suitable for the publication on this journal.

7. PLOS authors have the option to publish the peer review history of their article (what does this mean?). If published, this will include your full peer review and any attached files.

Reviewer #1: No

---

## [Editor Report · Acceptance letter]

2 Feb 2023

PONE-D-22-17370R2 

Metformin regulates myoblast differentiation through an AMPK-dependent mechanism 

Dear Dr. Maniscalco:

I'm pleased to inform you that your manuscript has been deemed suitable for publication in PLOS ONE. Congratulations! Your manuscript is now with our production department. 

Kind regards, 

on behalf of

Dr. Antonio Musaro 

Academic Editor

PLOS ONE